# Evaluation of an Innovative Point-of-Care Rapid Diagnostic Test for the Identification of Imported Malaria Parasites in China

**DOI:** 10.3390/tropicalmed8060296

**Published:** 2023-05-28

**Authors:** Kangming Lin, Shuqi Wang, Yuan Sui, Tao Zhang, Fei Luo, Feng Shi, Yingjun Qian, Jun Li, Shenning Lu, Chris Cotter, Duoquan Wang, Shizhu Li

**Affiliations:** 1Guangxi Zhuang Autonomous Region Center for Disease Control and Prevention, Nanning 530028, China; 2Anhui Provincial Center for Disease Control and Prevention, Hefei 230601, China; 3Brown School, Washington University, St. Louis, MO 63130, USA; 4Chongqing Center for Disease Control and Prevention, Chongqing 400042, China; 5National Institute of Parasitic Diseases, Chinese Center for Disease Control and Prevention (Chinese Center for Tropical Diseases Research), National Health Commission Key Laboratory of Parasite and Vector Biology, WHO Collaborating Center for Tropical Diseases, National Center for International Research on Tropical Diseases, Shanghai 200025, China; 6Malaria Elimination Initiative, Institute for Global Health Sciences, University of California San Francisco, San Francisco, CA 94109, USA; 7Department of Women’s and Children’s Health, Uppsala University, 75309 Uppsala, Sweden; 8School of Global Health, Chinese Center for Tropical Diseases Research, Shanghai Jiao Tong University School of Medicine, Shanghai 201100, China

**Keywords:** imported malaria, diagnosis, rapid diagnostic tests, post-elimination surveillance, China

## Abstract

Background: China was certified malaria-free by the World Health Organization on 30 June 2021. However, due to imported malaria, maintaining a malaria-free status in China is an ongoing challenge. There are critical gaps in the detection of imported malaria through the currently available tools, especially for non-*falciparum* malaria. In the study, a novel point-of-care Rapid Diagnostic Test designed for the detection of imported malaria infections was evaluated in the field. Methods: Suspected imported malaria cases reported from Guangxi and Anhui Provinces of China during 2018–2019 were enrolled to evaluate the novel RDTs. Diagnostic performance of the novel RDTs was evaluated based on its sensitivity, specificity, positive and negative predictive values, and Cohen’s kappa coefficient, using polymerase chain reaction as the gold standard. The Additive and absolute Net Reclassification Index were calculated to compare the diagnostic performance between the novel RDTs and Wondfo RDTs (control group). Results: A total of 602 samples were tested using the novel RDTs. Compared to the results of PCR, the novel RDTs presented sensitivity, specificity, PPV, NPV, and diagnostic accuracy rates of 78.37%, 95.05%, 94.70%, 79.59%, and 86.21%, respectively. Among the positive samples, the novel RDTs found 87.01%, 71.31%, 81.82%, and 61.54% of *P. falciparum*, *P. ovale*, *P. vivax*, and *P. malariae*, respectively. The ability to detect non-falciparum malaria did not differ significantly between the novel and Wondfo RDTs (control group). However, Wondfo RDTs can detect more *P. falciparum* cases than the novel RDTs (96.10% vs. 87.01%, *p* < 0.001). After the introduction of the novel RDTs, the value of the additive and absolute Net Reclassification Index is 1.83% and 1.33%, respectively. Conclusions: The novel RDTs demonstrated the ability to distinguish *P. ovale* and *P. malariae* from *P. vivax* which may help to improve the malaria post-elimination surveillance tools in China.

## 1. Introduction

Malaria remains a serious public health problem worldwide and is caused by *Plasmodium* parasites. It is reported that malaria cases were still on the rise between 2020 and 2021. However, the rate of increase is lower than that of 2019–2020; there were an estimated 247 million malaria cases in 2021 in 84 malaria endemic countries, this number was 245 million in 2020 and 232 million in 2019, and an estimated 619,000 malaria deaths [1], this number was 625,000 in 2020 and 568,000 in 2019. The African region accounts for 95% of the global malaria burden and 96% of malaria deaths. More seriously, during the COVID-19 pandemic, many factors, such as the stagnation of malaria prevention and control, the humanitarian crisis, the inadequacy of the health system, the shortage of funds, the biological threat, and the decline in the effectiveness of key disease control tools, such as drug impregnated mosquito nets, are hindering the realization of the goal of eliminating malaria globally. On the one hand, from 2000 to 2015, with the widespread application of malaria prevention and control interventions, the global incidence rate of malaria decreased by 27%, and the malaria mortality rate decreased by 50%. However, by 2017, the incidence rate had risen again, and the decline in the number of deaths had stalled [1]. on the other hand, in May 2015, the World Health Assembly released the global technical strategy for malaria 2016–2030, which set the most ambitious targets for malaria control and elimination thus far, namely reducing global malaria incidence and mortality rates by at least 90% by 2030 [2]. According to the requirements, by 2020, the incidence rate of malaria cases should be reduced by at least 40% and the mortality by at least 75%, but this key milestone goal has not been achieved [1]. Although the global decline in the malaria burden has stalled since 2015, 12 countries have been certified as malaria free since 2000 [1]. This includes China, which was certified malaria-free on June 30, 2021 by the World Health Organization (WHO) [3]. In addition, 13 countries reported zero indigenous cases for three consecutive years during this period.

With globalization and increased international movement, imported malaria cases continue to be reported in China, highlighting the challenges faced in preventing malaria re-establishment [4,5]. Prior to the COVID-19 pandemic, approximately 3000 imported malaria cases were reported each year, with Africa (89.1%) being the most common source [6]. These imported cases were mainly caused by overseas labourers [7], and the majority of infections were male (96.2%) [6]. In contrast, in some Western developed countries, the majority of cases are among individuals who contracted the infection while visiting friends and relatives [8,9]. Interestingly, recent evidence has shown that the proportion of imported malaria cases caused by non-*falciparum* malaria, especially *Plasmodium ovale* (*P. ovale*), increased to levels higher than expected [10]. Moreover, the proportion peaked at nearly 15% in 2018 in China [11].

Anhui Province is located in the southeastern part of China, in the middle and lower reaches of the Yangtze and Huai Rivers, with an area of 140,100 km^2^ and a land area of 139,400 km^2^. It is a transitional region between warm temperate and subtropical climates, with a distinct monsoon climate. Anhui Province was once one of the key malaria endemic provinces in China, which seriously affected people’s physical health and socio-economic development. Anhui Province is an unstable malaria endemic area, and historically, the Huaibei Plain was an endemic area for *P. vivax*; in the hilly areas of the Jianghuai River, there are many cases of daily malaria, and some areas have the presence of *P. falciparum*; the mountainous areas in southern Anhui are mainly characterized by *P. vivax*. Prior to the 1970s, there were cases of *P. falciparum* and a small number of *P. malariae*. After nearly 70 years of prevention and control, the basic elimination of *P. falciparum* was achieved in 1996. The last local infection case in the province was reported in 2013. Since 2014, there have been no local cases of malaria reported in the province, and all cases have been imported cases. Among them, there were 68–190 reported cases from 2011 to 2019, mainly of *P. falciparum*, with reports of the other three species and mixed infections [12].

Guangxi is located in the southern part of China, bordering Guangdong and Hunan in the southeast and northeast, Yunnan and Vietnam in the west and southwest, and Guizhou Province in the north. The average annual temperature is 16.5–23.1 °C. There are 14 cities and 111 counties in the province, with a total area of 236,000 km and a total population of 56.95 million in 2019. Throughout history, Guangxi has been a severely prevalent area for malaria, with major outbreaks occurring in 1954, 1963, and 1971. Throughout history, malaria in Guangxi was mainly caused by *P. vivax* and *P. falciparum*. After the founding of New China, after more than 70 years of prevention and control, the incidence rate dropped from 296.7/10,000 in 1954 to less than 1/10,000 in 1987; The last local infection case was reported in 2012, and since 2013 there have been no local infection cases reported in the province. All cases occurred as imported cases, mainly from African and Southeast Asian countries and regions. Among them, a total of 3195 malaria cases were reported in Guangxi from 2010 to 2019, with the main species being *P. falciparum*, while the other three species and mixed infections were all reported as well [13,14].

In China, when suspected malaria patients presenting with symptoms of malaria, especially combined with a history of travel to a malaria-endemic area, seek medical care, the physician provides a diagnostic test for malaria, commonly microscopy or rapid diagnostic test (RDTs) [15]. However, due to their morphological similarity, *P. ovale* is easily and commonly misdiagnosed as an infection of *P. vivax* [16,17], which may lead to an inappropriate case management treatment response. For someone self-diagnosing for malaria, RDTs are available in Chinese pharmacies; however, these also cannot differentiate between *P. malariae*, *P. ovale*, and *P. vivax* infections. Thus, prompt and precise diagnostic tools that can detect and differentiate non-*falciparum* malaria species are needed.

In this study, a novel point-of-care RDT was designed for the detection of imported malaria infections and was evaluated using polymerase chain reaction (PCR) as the gold standard. We believe that the newly designed diagnostic tool can benefit the prevention of malaria re-establishment in the future.

## 2. Materials and Methods

### 2.1. Study Setting, Participants and Design

The provinces of Anhui and the Guangxi Autonomous Region were selected as study areas for the evaluation of the novel RDTs (Figure 1). Historically, malaria was highly prevalent in Anhui province [12], which presents a high risk of re-establishment of malaria. Guangxi is a border province in southern China that exports large numbers of migrant workers to Africa. Since 2013, the number of imported malaria cases in Guangxi Province has been among the highest in China [5].

In China, each suspected malaria case should be mandatorily reported through the China Information System for Disease Control and Prevention (CISDCP) [18]. This is a real-world study based on the surveillance system of malaria in China. All suspected imported malaria patients reported in Anhui and Guangxi provinces from 2018 to 2019 were enrolled as participants in the study. Individuals were contacted by telephone to obtain verbal informed consent. An imported case was defined as a malaria infection acquired outside the country (in this study, China).

In China, when a suspected case is reported through the network, the blood samples, including whole blood and smears that were collected from the patient before anti-malarial treatment [19], are sent to the provincial reference laboratory for final confirmation using microscopic examination and polymerase chain (PCR) reaction according to the malaria diagnostic criteria in China [20]. In this study, the real-time PCR method, used in the form of commercial real-time PCR Kits (Shanghai ZJ Bio-tech Co., Ltd., Shanghai, China), was taken as the gold standard to detect the malaria infection and further distinguish between *Plasmodium* species. The kits, targeting the 18s rRNA gene, were designed by referring to a previous study and provided internal control [21]. Before PCR test, DNA was extracted using the QIAamp DNA Mini kit (QIAGEN Inc., Hilden, Germany) in accordance with the manufacturer’s instructions. Then, PCR was performed in a 40.4-μL reaction mixture containing 35 μL reaction mix, 0.4 μL enzyme mix, 1 μL internal control, and 4 μL DNA template. The reaction conditions were as follows: 37 °C for 2 min and 94 °C for 2 min, followed by 40 cycles at 93 °C for 15 s and 60 °C for 60 s. On the other hand, a commercial test strip (Diagnostic Kit for Malaria, Guangzhou Wondfo Biotech Co., Ltd., Guangzhou, China) detecting Pf-HRP2 (Human histidine rich protein 2) and Pan- lactate dehydrogenase (LDH) was taken as control in comparison to the novel RDTs in the laboratory setting. An imported case was defined as a malaria infection acquired outside the country (in this study, China).

### 2.2. Interpretation of the Results for RDTs

A novel malaria RDT was designed by the National Institute of Parasitic Diseases, Chinese Center for Disease Control and Prevention and tested in this study. It is not yet officially available commercially. The novel RDT (Figure 2) is an immunochromatographic test strip, and has one control line and three test lines (“T1”, “T2”, and “T3”), detecting Pf-HRP2, Pv-speciifc LDH, and Pan-LDH, respectively. If the infection was caused by *P. vivax*, T2 and T3 line were simultaneously positive, whereas, if the infections were *P. ovale* and/or *P. malariae,* only the “T3” line was positive, with a negative “T2” line. Using the combination of “T2” and “T3” test lines, the novel mRDT can distinguish *P. vivax* from *P. ovale* and/or *P. malariae* (Table 1). Wondfo RDTs have one control line (“C”) and two detection lines (“T1” and “T2”). Additionally, the T1 and T2 lines indicate *P. falciparum* and *Plasmodium* infections, respectively. Blood samples from participants were tested simultaneously with novel and Wondfo RDTs in the provincial laboratory reference as directed by the manufacturer.

### 2.3. Data Analysis

The categorical data are presented as percentages. Values are presented as the mean ± standard deviation for data that were normally distributed. Differences in proportions were compared using McNemar’s χ2 test. Taking the results of PCR as the gold standard, the diagnostic performances of the novel and Wondfo RDTs were presented with the following parameters: sensitivity, specificity, positive (PPV) and negative predictive values (NPV), and Cohen’s kappa coefficient, with their respective 95% confidence intervals. The formula for calculating PPV and NPV is: PPV = (true positives)/(true positives + false positives), NPV = (true negatives)/(true negatives + false negatives), respectively. Additive Net Reclassification Index (NRI) and absolute NRI are calculated to compare diagnostic performance between the Novel and Wondfo RDTs [22]. All statistical tests were two-sided, and *p* < 0.05 was considered statistically significant. The study data were recorded and entered into an Excel database (Microsoft Corporation, Redmond, WA, USA), and analysis was performed using SPSS 26.0 statistical software (SPSS Inc., Chicago, IL, USA). The thematic map of geographic distribution was created by MapInfo 15.0 (Pitney Bowes Inc., Troy, NY, USA).

## 3. Results

During the study period, a total of 602 blood samples collected from suspected malaria cases were tested to evaluate the performance of the novel and Wondfo RDTs. Cases came from 26 African and 2 Asian countries, with Africa (600; 99.67%) being the most common region of origin. The five countries of origin of infection were Ghana (115; 19.10%), Nigeria (55; 9.14%), Ivory Coast (53; 8.80%), Angola (51; 8.47%), and Mozambique (50; 8.31). Of these, 154 (*P. falciparum*), 123 (*P. ovale*), 22 (*P. vivax*), 13 (*P. malariae*), and 7 (mixed infections) samples tested positive. The remaining 283 cases were confirmed negative by PCR. The mean age of the participants was 42.2 ± 9.1 years, and 578 participants (96.0%) were males.

### 3.1. Diagnostic Performance of the Novel and Wondfo RDTs

Compared to the results of PCR, the novel RDTs presented sensitivity, specificity, PPV, NPV, and diagnostic accuracy rates of 78.37%, 95.05%, 94.70%, 79.59%, and 86.21%, respectively. Those of the Wondfo RDTs were 86.21%, 89.05%, 89.87%, 85.14%, and 87.54%, respectively. In terms of sensitivity, Wondfo RDTs outperformed the novel RDTs (86.21% vs. 78.37%), whereas the opposite is true for specificity (89.05% vs. 95.05%) (Table 2).

Both RDTs were able to detect all four *Plasmodium* species from the blood samples which were collected. Compared to the PCR gold standard, the Wondfo RDTs detected 96.01% (*P. falciparum),* 72.13% (*P. ovale*), 90.91%(*P. vivax*), and 92.31%(*P. malariae*), while the novel RDTs identified 87.01% (*P. falciparum*), 71.31%(*P. ovale*), 81.82%(*P. vivax*), and 61.54% (*P. malariae*) of cases. Their ability to detect non-*falciparum* malaria did not differ significantly, but Wondfo RDTs detected more *P. falciparum* infections than the novel RDTs (96.10% vs. 87.01%) (Table 3).

### 3.2. Additive NRI and Absolute NRI

The additive NRI and absolute NRI were calculated to assess the improvement due to the novel RDTs introduced in the field, compared to Wondfo RDTs. The values of the additive NRI and absolute NRI are 1.83% and 1.33%, respectively. The results showed that there was no difference in diagnostic ability between the Novel and Wondfo RDTs (all *p* > 0.05) (Table 4).

## 4. Discussion

The last indigenous malaria case in China was reported in 2016 and local transmission has been interrupted since 2017 [23]. China was certified malaria-free by the WHO in 2021 and is facing continued challenges due to imported malaria, particularly among male workers visiting Africa. Therefore, performing and sustaining a sensitive surveillance system that can detect suspected malaria cases in a prompt and accurate manner is the key. However, the inability to properly detect and distinguish malaria parasites is a huge barrier [24]. In field practice, microscopy and RDTs are common diagnosis methods for malaria in the health care setting in China. However, sustaining microscopy competency is extremely difficult due to the limited accumulation of experience [25]. RDT has the characteristics of easy operation and intuitive reading; therefore, it is the diagnostic method for malaria that has been recommended by the WHO. Further, RDTs, a vital supplement, extend access to diagnostic tools in areas where microscopy cannot be reliably maintained. RDT has been recommended to provide parasite diagnosis for suspected malaria cases by WHO [26].

For malaria detection, RDT that can distinguish between the types of malaria parasites, live and dead infections, and the sexual stage of parasitemia will be very helpful for diagnosis and guide intervention measures. The main challenge is in the field of low-level parasitemia. The examination of the life cycle stage of malaria parasite infection has identified many key targets, including the HRP2 protein (*P. falciparum*), parasite lactate dehydrogenase (pLDH, *Plasmodium* genus), and malaria parasite aldolase (*Plasmodium* genus). Therefore, distinguishing between *P. falciparum* and other species is not a simple task [27]. Piper conducted research on how existing combinations of pLDH antibodies perform in the differential diagnosis of *P. falciparum*, pan specificity, and malaria parasites, showing that differences in reactivity may be related to small differences on the surface of pLDH, with subtle amino acid changes being the cause of species specificity [28].

There is evidence to suggest that, in countries with low malaria transmission, due to the long-term absence of malaria cases, the awareness and vigilance of health systems and the preparedness of health workers towards the correct management of suspected malaria will decrease [29].

In China, infections caused by *P. falciparum*, *P. vivax*, *P. malariae*, and *P. ovale* have still been reported for many years. The malaria surveillance system needs to introduce RDTs with the ability to detect four species. However, thus far, only two Pf/Pan tests (Wondfo and BinaxNOW^®^ Malaria) have been registered in the National Medical Products Administration that could be used in health facilities. Wondfo RDTs are used more in the market for price reasons, and presented a better performance for detecting *P. ovale* compared to CareStart pLDH PAN and SD BIOLINE Pf/Pan RDTs [30]. According to the manufacturer’s instructions, Wonfo RDTs can distinguish *P. falciparum* from *non-falciparum* species. However, they could not further differentiate *non-falciparum* species. *P. vivax*, a common species in all malaria-endemic areas, is distributed nationwide and is the main species related to the risk of malaria reintroduction. Due to their morphological similarity, *P. ovale* is easily and commonly misdiagnosed as an infection of *P. vivax* in the field, further leading to inappropriate interventions. A novel RDT has been designed to fulfil the gap in field practice.

Related studies have shown that, although the protein sequence of pLDH is very conservative within the same species, there are certain differences in the protein sequence of pLDH among the four human malaria parasites. Usually, the detection antibodies in RDT are monoclonal antibodies that are prepared based on a specific pLDH antigen of a certain insect species. Therefore, when detecting unknown samples, if the patient is infected with other insect species in the body, the detection antibodies in RDT cannot specifically bind to the antigen in the sample, resulting in false negative test results. The specificity experiment results showed that the monoclonal antibody of PfLDH only reacted with the samples of *P. falciparum* and did not react with the samples of *P. vivax* (P. vi-vax); similarly, PvLDH antibodies only react with *P. vivax* samples and do not react with *P. falciparum* samples [31].

In this study, the diagnostic performance of the novel RDTs was assessed using blood samples collected from the field. The novel RDTs found 87.01% of *P. falciparum*, 71.31% of *P. ovale*, 81.82% of *P. vivax*, and 61.54% of *P. malariae* infections. The diagnostic performance of the novel RDTs for *non-falciparum* species was similar to that of the Wondfo RDTs. Importantly, the novel RDTs had the ability to distinguish *P. ovale* and *P. malariae* infections from *P. vivax*. Although improvements are still needed to improve the novel RDTs, we believe that significant progress has been made in this initial development and through this diagnostic evaluation.

Although the diagnostic sensitivity of the novel RDTs for *P. falciparum* was lower than for Wondfo RDTs (87.01% vs. 96.10%), *P. falciparum* is the dominant species of imported malaria [4]. Novel RDTs need to fulfil the gap in future practical applications. The value of additive NRI and absolute NRI are 1.83% and 1.33%, respectively, which means that there are no obvious additional benefits to the introduction of the novel RDTs in the field.

*P. ovale* and *P. malariae* were commonly considered as the ‘bashful’ malaria parasites, due to their low prevalence and limited geographic distribution [32]. However, the introduction of more sensitive molecular methods has provided more evidence that their geographic distribution is larger than previously speculated [33,34]. Further, scientific knowledge about the two species is very limited compared to *P. falciparum* and *P. vivax*. Developing diagnostic tools that target the two main species of *Plasmodium* is challenging. One factor significantly impacting its success is that parasitaemia is typically very low in infections caused by *P. ovale* and *P. malariae* (sub-microscopic malaria infections). According to a previous study, *P. malariae* only invades aged red blood cells (0.1% parasitaemia) and *P. ovale* preferentially invades youthful red blood cells (1% parasitaemia) [35]. This implies that the concentration of specific antigens targeted by novel RDTs may be lower than the threshold value. For example, in our field assessment, novel RDTs only found 61.54% infections of *P. malariae*.

Our study has two main limitations. First, limitations around clinical and patient-level RDTs occur frequently in non-endemic settings, especially in the presence of treatment delays [36]. Therefore, the performance of the novel RDTs may be underestimated. Second, the number of *P. malariae* and *P. vivax* cases is small.

## 5. Conclusions

The diagnostic power of novel RDTs for non-falciparum species detection was comparable to the Wondfo RDTs, and demonstrated the ability to distinguish *P. ovale* and *P. malariae* from *P. vivax*. We believe that the newly designed RDTs can benefit POR practices within China and other malaria-eliminating and POR countries with further improvements.

## Figures and Tables

**Figure 1 tropicalmed-08-00296-f001:**
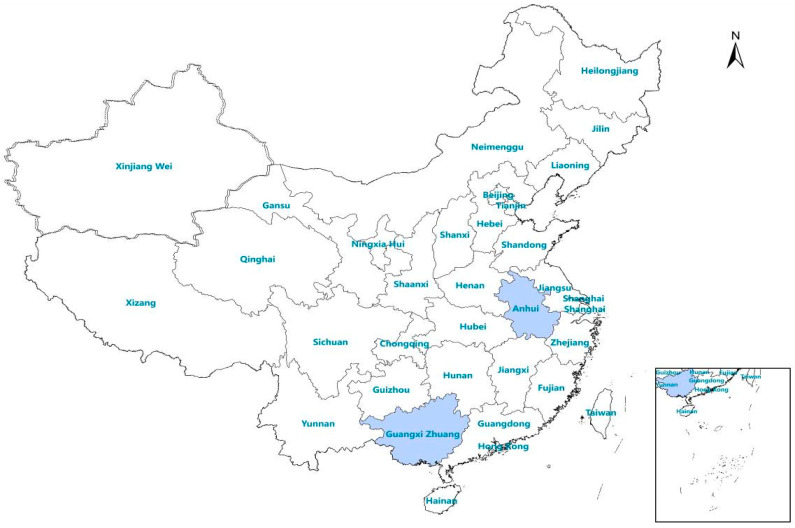
Location of the study area in China.

**Figure 2 tropicalmed-08-00296-f002:**
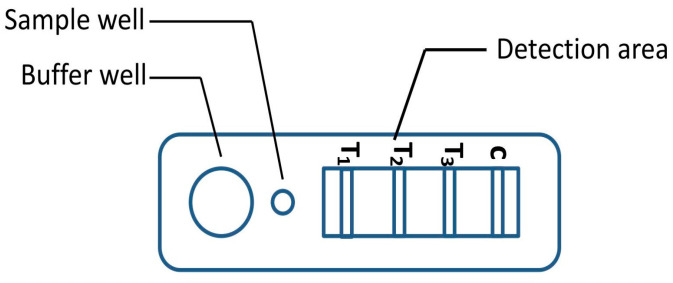
Schematic illustration of the novel malaria RDTs.

**Table 1 tropicalmed-08-00296-t001:** Interpretation of the results of the novel malaria RDTs.

Valid	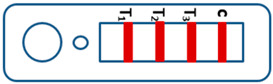	Mixed infection of *P. falciparum, P. vivax* and *P. ovale* or/and *P. malariae*	Invalid	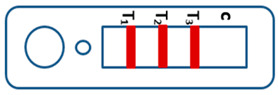
	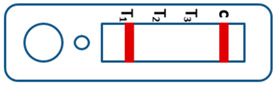	Positive result for *P. falciparum* only		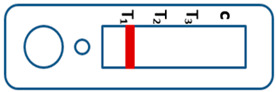
	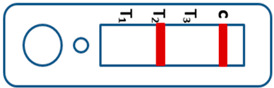	Positive result for *P. vivax* only		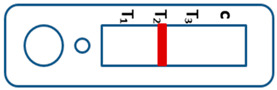
	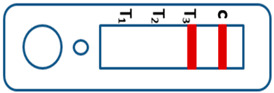	Positive result for *P. ovale* and/or *P. malariae*		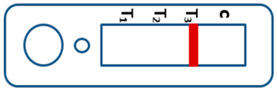
	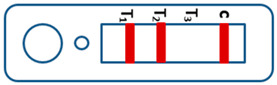	Mixed infection of *P. falciparum* and *P. vivax*		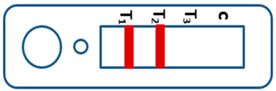
	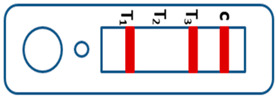	Mixed infection of *P. falciparum* and *P. ovale* and/or *P. malariae*		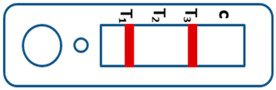
	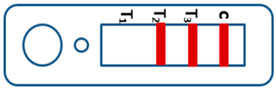	Mixed infection of *P. vivax* and *P. ovale* and/or *P. malariae*		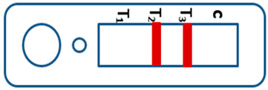
	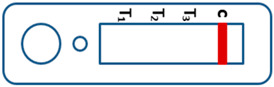	Negative for any *Plasmodium*		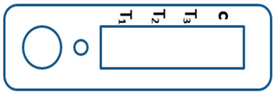

**Table 2 tropicalmed-08-00296-t002:** Diagnostic performance of Novel and Wondfo RDTs for malaria in a laboratory setting.

Characteristics	The Novel RDTs	Wondfo RDTs	χ^2^ *	*p*-Value
Sensitivity [95% CI]	78.37 [73.83–82.91]	86.21 [82.40–90.01]	11.294 *	0.001
Specificity [95% CI]	95.05 [92.51–97.59]	89.05 [85.39–92.71]	13.474 *	<0.001
PPV [95% CI]	94.70 [91.98–97.42]	89.87 [86.47–93.27]	4.543	0.033
NPV [95% CI]	79.59 [75.27–83.90]	85.14 [81.06–89.21]	3.318	0.069
Diagnostic accuracy rate [95% CI]	86.21 [83.45–88.97]	87.54 [84.90–90.19]	0.466	0.495
Kappa value [95% CI]	0.726 [0.779–0.673]	0.751 [0.698–0.804]	NA	NA

PPV: positive predictive value, NPV: negative predictive value, CI: confidence interval, NA, not applicable * McNemar’s χ^2^ test.

**Table 3 tropicalmed-08-00296-t003:** Detection ability of novel and Wondfo RDTs for different malaria species in a laboratory setting.

Species *	Type of RDTs	N	Results of RDTs (*n*)		Sensitivity (%)	χ^2^ #	*p*-Value
			Positive	Negative			
*P. falciparum*	The novel	154	134	20	87.01	12.071	<0.001
	Wondfo	154	148	6	96.10		
*P. ovale*	The novel	123	87	35	71.31	0.036	0.850
	Wondfo	123	88	34	72.13		
*P. vivax*	The novel	22	18	4	81.82	0.500	0.500
	Wondfo	22	20	2	90.91		
*P. malariae*	The novel	13	8	5	61.54	2.250	0.134
	Wondfo	13	12	1	92.31		

* Species identification results provided by the Anhui and Guangxi Malaria Diagnostic Reference Laboratory. # McNemar’s χ^2^ test.

**Table 4 tropicalmed-08-00296-t004:** Comparison of diagnostic ability between the Novel and Wondfo RDTs.

Positive Samples (n = 309)				
		The Novel RDTs		
Wondfo RDTs		Negative	Positive	Total
	Negative	31	38	69
	Positive	13	237	250
Negative samples (n = 283)				
		The novel RDTs		
Wondfo RDTs		Negative	Positive	Total
	Negative	251	18	269
	Positive	1	13	14
	Additive NRI	1.83%	Absolute NRI	1.33%
	Z	0.673	Z	1.317
	*p*	0.501	*p*	0.188

## Data Availability

The dataset analysed during the current study are available from the corresponding authors on reasonable request.

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
