# Peer review of "Evaluation of an Innovative Point-of-Care Rapid Diagnostic Test for the Identification of Imported Malaria Parasites in China"

_tropicalmed, 2023, doi:10.3390/tropicalmed8060296_

Round 1
Reviewer 1 Report
The paper, although describes a valid concern, lacks scientific merit in design and reporting. There are major flaws in interpreting the results with respect to the detection and identification of P. ovale and P. malariae.
Author Response
Revision note and responses
Response to reviewer 1’s comments
Comment 1
The paper, although describes a valid concern, lacks scientific merit in design and reporting. There are major flaws in interpreting the results with respect to the detection and identification of P. ovale and P. malariae.
Response:
​Thanks for your comments. We agree with you. In this study, we put the use of originally designed mRDT into a real-world situation in comparison with those from the routine diagnostic tools in terms of sensitivity, specificity, PPV, NPV, and diagnostic accuracy rate (operational research) through the NMEP in the study areas.
All the suspected malaria cases reported in Anhui and Guangxi Provinces through the China Information System for Disease Control and Prevention were enrolled during the study period. Diagnostic performance as well as limitations of the novel mRDT were initially evaluated and we believe that these findings will help to further improve the performance of novel mRDT which will help improve the malaria post-elimination surveillance in China, particularly for the potential identification of P. ovale and P. malariae.
The interpretation of the test results for the novel mRDT is the same as other commercial kits such as the CarestartTM Malaria Pf/Pan Ag Combo RDT. Compared to
CarestartTM Malaria Pf/Pan Ag Combo RDT, we added an additional test line to detect Pv-speciifc LDH in the novel mRDT. Therefore, it can further distinguish P. vivax from P. ovale and P. malariae. In China, P. vivax was the historically most widely distributed and mainly related to the risk of malaria reintroduction from imported cases. ​Due to their morphological similarity through the microscopy, P. ovale is often easily misdiagnosed as an infection of P. vivax in the field implementation, leading to the inappropriate interventions. Therefore, we designed a novel mRDT to try to fulfill the needing gap in surveillance of imported malaria in China. On the other hand, the results of malaria parasite species confirmation were based in the provincial reference laboratory as required from NMEP. So, we think that interpretation of the results with respect to the detection and identification of P. ovale and P. malariae will be much useful and necessary for the malaria post-elimination surveillance in China.

Reviewer 2 Report
The study is of significant importance since a new mRDT was evaluated. However, minor revision is required as the manuscript is not publishable in its current form.
Lines 24-29: The details of the novel mRDT were missing that is brand name, country of origin.
Lines 58-59: These infectees were asymptomatic so the use of patients may not be appropriate.
Line 79: The Study setting and participants were well described however, the study design was missing. Study design means something specifically in research work. Include the specific design employed in this study.
Line 109: Didn’t the novel mRDT has a control line?
Line 125: include how the Additive Net Reclassification Index (NRI) and Absolute NRI were calculated in the data analysis section
Include the following subheadings under the Material and methods section
i. The nature of the samples you collected. Were they DBS, frozen or refrigerated samples
ii. How you obtained DNA for the real-time PCR
iii. Since the novel mRDT was designed by the National Institute of Parasitic Diseases, Chinese Center for Disease Control and Prevention, instructions for use is unavailable. Please include
iv. Is the novel mRDT commercially available?
v. Indicate how the samples were collected. Was it systematically, randomly or by confidence?
vi. Was the malaria status of these samples known, by any other means?
Line 134: I think including the geographical locations of these imported cases will be good.
Lines 135-6: None of the RDTs could differentiate Po from Pm, by what means did you determine these: 123 (P. ovale) and 13 (P. malariae)
Additional comment: Can authors reevaluate the mRDT with respect to malaria parasitemia in the blood samples.
Reviewer 3 Report
Lin et al developed and tested a novel Rapid Diagnostic Test for the detection of malaria parasites in their MS. Despite the well-written manuscript, there are critical scientific and technical concerns that mislead the results and thus the conclusions of this MS.
General comments
Given that there are no RDTs that specifically target P. ovale and/or P. malariae, the novel RDT developed in this study is very exciting. However, fundamental questions must be answered before the authors conclude that the novel RDTs can distinguish P. ovale and P. malariae infections from P. vivax infection.
1. The authors stated in lines 108 -110 that (The novel RDT (Figure 2) is an immunochromatographic test strip with three test lines ("T1", "T2", and "T3") that detect Pf-HRP2, Pv-specific LDH, and Pan-LDH, respectively (Table 1). Given that all current RDTs use Pan-LDH to identify non-falciparum malaria (Pv, Pm, and Po), it is unclear how line T3 distinguishes Pv from Pm and/or Po. Thus, how Pan-LDH in T3 of the novel develops RDT distinguishes Pv from other species must be discussed in detail.
2. Despite the fact that microscopy is the gold standard test for malaria, the authors used PCR as the gold standard. Why was microscopy not used, what were the PCR target genes, and how were plasmodium species identified by PCR?
3. I believe the standard protocol for evaluating any diagnostic test was not fully followed in this study; if this is not the case, the process by which the novel RDT was developed should be described in detail. Furthermore, the cross-reactivity of the novel RDT with samples positive for other infections was not evaluated.
4. Given that the blood samples were collected in 2018-2019, I'm not sure if the study's findings should be re-evaluated or not, given the genetic diversity of malaria parasites.
Specific comments
Title
1. The title does not accurately represent the manuscript's procedure and findings.
A. The term development should refer to a specific test development procedure that is not covered in the manuscript.
B. The title includes the phrase (the identification of imported malaria parasites in China), and whether there is a difference in the detection of imported and non-imported malaria, or how they developed RDT will do that is not covered by this manuscript.
Materials and Methods
1. Line 79-105: Study setting, participants, and design ( the authors didn’t mention any information about the participants
2. Although the main objective of this study is to develop and evaluate a novel RDT, the authors only include the Interpretation of the results for RDTs section. A section describing how the novel RDT was developed should be included
3. The authors need to scientifically clarify how Pan-LDH was used as a marker for P. ovale and/or P. malariae. Also, justify how the novel RDT differentiates between non-falciparum while the Wondfo RDTs do not, despite the fact that both use Pan-LDH.
4. For the PCR protocol, the authors cited Reference 18, which should be checked.
Results
1. Table 3 shows the detection ability of novel and Wondfo RDTs for different malaria species Given that Wondfo RDTs only have two detection lines for Plasmodium falciparum and other Plasmodium infections, it is unclear how the authors obtained the species-specific numbers in table 3 (Po = 123, Pv = 22, and Pm = 13) using Wondfo RDT. I believe the data in Table 3 was incorrect.
Reviewer 4 Report
The idea of the present study is of major interest for the fight against malaria in the sense that it seeks to improve the diagnostic methods that can be decisive in the context of pre-elimination and elimination of the disease.
Major remarks
The title of the manuscript does not seem to reflect its content in view of the methodology and results presented.
The authors put a major emphasis on the innovative approach of their work, however, compared to what exists, this does not seem to be the case as there are rapid tests available to detect different species of Plasmodium. Moreover, in the interpretation of their results, the new test created is less efficient than the existing one.
The PCR technique used should be sufficiently described: target genes, primer and probe sequences. The same should be done for the internal control
The volume indicated for the pcr reaction mix does not correspond to the sum of the volumes of all pcr reagents
Concerning the rapid diagnostic system, apart from the targeted antigens, there is no information about the principle and the mechanism of operation.
Furthermore, the authors should explain how they came to interpret the fact that Pan-LDH is specific to malaria and or oval? How is it possible to detect Pan-LDH without testing positive for P. falciparum or P. vivax?
In the methodology used, it would be beneficial to see the reason(s) for the relatively low sensitivity of their new test compared to the existing one. Is this related to parasite density or other parameters? This question deserves to be explored, especially since the authors have done PCR and microscopy.
Minor remarks
The formulas used to calculate the positive and negative predictive values should be included in the methodology.
In the presentation of Table 1, it should indicate on the left "valid" as what is done right to facilitate the understanding of the table
